# Trends in Women’s Empowerment and Their Association with Childhood Vaccination in Cambodia: Evidence from Demographic and Health Surveys (2010–2022)

**DOI:** 10.3390/vaccines14010048

**Published:** 2025-12-31

**Authors:** Haizhu Song, Yanqin Zhang, Qian Long

**Affiliations:** 1Global Health Research Center, Duke Kunshan University, No. 8 Duke Avenue, Kunshan 215316, China; hs372@duke.edu (H.S.); carriezhyq@163.com (Y.Z.); 2Duke Global Health Institute, Duke University, Durham NC 27708, USA

**Keywords:** women’s empowerment, child immunization, child vaccine, public health, Cambodia, national immunization program, health policy

## Abstract

**Background**: Women’s empowerment has been significantly associated with improved child health outcomes. Cambodia, amid a rapid socioeconomic transition, offers a critical setting to examine how advancements in women’s empowerment over the past decade have influenced child immunization completion within the first two years of life. **Methods**: Data from the Cambodia Demographic and Health Surveys conducted in 2010, 2014, and 2021–22, encompassing 9222 women with recent births, were analyzed. Empowerment was measured across literacy and information access, employment, and decision-making domains. Multinomial logistic regression assessed associations between empowerment factors and completion of oral polio (OPV), diphtheria–tetanus–pertussis (DTP), pneumococcal conjugate (PCV), and measles–rubella (MR) vaccines, adjusting for demographic and socioeconomic variables. **Results**: Between 2010 and 2022, women’s empowerment in Cambodia improved significantly, marked by higher literacy rates, nearly half of women completing primary education, and expanded digital access, with 82.4% owning mobile phones and approximately 50% using the internet daily. While non-working women slightly increased, agricultural employment declined by 20%, and cash earnings rose from 48.7% to 82.5%. Most women participated in major household decision-making, either independently or jointly. Completion rates for OPV, DTP, and PCV ranged from 79% to 83%, while just over half of children were fully vaccinated against measles. Higher maternal education and cash earnings were positively associated with OPV, DTP, and PCV completion but negatively associated with measles vaccination. Women in agricultural work were less likely to complete measles vaccination for their children than non-working women. Joint decision-making regarding the use of respondents’ income was associated with a higher likelihood of measles non-completion (OR = 2.26, 95% CI: 1.13–4.51), whereas joint decision-making about respondents’ health care was associated with a higher likelihood of measles completion (OR = 0.42, 95% CI: 0.21–0.83). **Conclusions**: Women’s empowerment remains a key determinant of vaccination outcomes in Cambodia. The distinct pattern observed for measles suggests that vaccines scheduled for older ages encounter greater structural and behavioral barriers. To overcome these challenges, strategies should focus on enhancing defaulter tracking, implementing reminder systems, expanding outreach and catch-up programs, and improving the convenience of vaccination services.

## 1. Introduction

Women’s empowerment is a multidimensional and dynamic concept, commonly defined as the process of enhancing women’s ability to make strategic life choices and exercise control over their own lives [1,2]. In practice, empowerment encompasses women’s access to enabling resources, autonomy in decision-making, and the ability to achieve desired outcomes across various life domains [3]. Beyond its inherent significance as a fundamental human right, women’s empowerment is increasingly recognized as a development priority. It has been established as a standalone goal under the Sustainable Development Goals (SDG 5: Achieve gender equality and empower all women and girls) and is widely recognized as essential for fostering sustainable development and enhancing societal well-being [4].

Over the past two to three decades, low- and middle-income countries (LMICs) have made notable progress in advancing women’s empowerment, particularly by expanding access to education, increasing female labor force participation, and improving access to health services [5]. Women’s empowerment, in turn, has a profound impact on both family and child health outcomes. Numerous empirical studies have shown that, in LMICs, women with higher levels of empowerment are more likely to actively seek and use child health services, such as ensuring their children receive timely immunizations, complete the full vaccination schedule, and reduce the risk of preventable diseases. Dimensions of women’s empowerment, particularly decision-making autonomy and freedom of movement, have generally been associated with higher odds of childhood immunization, although the strength and direction of these associations vary across contexts [6]. However, in some LMIC settings, these pathways may not function uniformly across vaccine schedules. Completion of early multi-dose infant vaccines is often driven by initial contact with health services, whereas dropout between early doses and later-scheduled vaccines is commonly seen as an indicator of persistent access barriers and challenges in maintaining engagement beyond infancy [7]. Despite these advancements, the process of achieving full empowerment remains hindered by deeply entrenched social norms, restrictive gender roles, and structural inequalities. These enduring challenges continue to limit the complete realization of women’s empowerment.

Cambodia transitioned from a low-income country to a lower-middle-income country in 2015, with a reported GDP per capita of USD 2754.5 in 2024 [8]. In the same year, Cambodia’s total population was approximately 17.6 million [9], of which about 4.8 million were women aged 15–49 [10]. The sex ratio was roughly 49:51, indicating slightly fewer men than women [11]. Over the past decade, although gender inequality has narrowed, the United Nations Development Program’s Gender Inequality Index (GII) still reported a score of 0.461 for Cambodia, ranking around 116th among more than 160 countries worldwide, indicating persistent gender disparities [12]. In terms of educational attainment, Cambodian women have made significant progress, as evidenced by an increase in the adult female literacy rate from approximately 75% in 2015 to 79.7% in 2022, although it remains below the male rate of 88% [13]. Given employment opportunities, women’s labor force participation has remained at around 75%, close to that of men, but they are predominantly concentrated in low-wage, informal sectors, with average earnings only about 80% of men’s [14].

In Cambodia, the under-five mortality rate declined markedly from about 124 per 1000 live births in 2000 to 16 per 1000 in 2021–22 [15], a reduction related mainly to the expansion of the National Immunization Program (NIP, formerly the Expanded Program on Immunization, EPI) [16]. Since its inception in 1974, Cambodia’s national immunization program, initiated by the World Health Organization (WHO), has targeted six basic vaccines: diphtheria, tetanus, pertussis, measles, poliomyelitis, and tuberculosis. Over time, the program has expanded to include 13 vaccines recommended by WHO, such as hepatitis B (HepB) and the pneumococcal conjugate vaccine (PCV), to address broader child health needs [17]. This expansion has been successfully facilitated by Gavi, the Vaccine Alliance, which has provided critical support since 2002 through vaccine procurement and health system strengthening [18]. As a result, Cambodia has made significant progress in achieving high vaccination coverage: according to WHO/UNICEF 2023 estimates, completion of the third dose of the diphtheria-tetanus-pertussis vaccine (DTP3) reached 85%, while Bacillus Calmette–Guérin (BCG) coverage was approximately 91% [19]. Overall, the combined efforts to expand vaccine coverage and strengthen health systems reflect substantial progress in safeguarding child health in Cambodia.

Few studies have examined the impact of women’s empowerment on child health in Cambodia. This study examines trends in key women’s empowerment dimensions from 2010 to 2021–22, including literacy and information retrieval, employment, and participation in household decision-making, and assesses how these factors are associated with children’s completion of core vaccinations. By analyzing these empowerment domains, the study provides empirical evidence on the extent to which women’s autonomy and access to resources shape the uptake of essential preventive health services, thereby contributing to improved child health outcomes.

## 2. Materials and Methods

### 2.1. Data Sources and Study Population

This study analyzed data from the Cambodia Demographic and Health Surveys (CDHS) conducted in 2010, 2014, and 2021–22. These surveys were jointly implemented by the Cambodian National Institute of Statistics (NIS) and the Ministry of Health (MoH). The CDHS employed a two-stage stratified sampling design. In the first stage, sample points were selected from census tracts defined in the General Population Census [20], followed by random selection of households within each cluster. This approach ensured national and regional representativeness. All women aged 15 to 49 years, whether permanent residents or temporary visitors who had stayed in the sampled households the night before the survey, were eligible for interview. Detailed information on the survey methodology is available in the respective CDHS technical documentation.

For this study, the analytic sample included women who had a live birth within the two years preceding each survey. In cases where a respondent reported multiple births during this period, only the youngest child was included in the analysis to avoid overrepresentation of women with higher fertility.

### 2.2. Measure of Women’s Empowerment

Women’s empowerment was measured using CDHS questionnaire items in three domains: (1) Literacy and information retrieval, (2) Employment, and (3) Decision-making. These domains and their indicators were selected according to DHS conventions for gender-related variables. Women’s empowerment indicators were analyzed separately rather than as a composite construct, as different dimensions may operate through distinct pathways and show divergent associations with vaccination outcomes. Specifically:

#### 2.2.1. Literacy and Information Retrieval

This dimension was assessed using indicators of formal education, reading ability, media exposure, and digital access. Educational attainment was categorized as no education, incomplete primary, primary, secondary, or higher, based on questions about school attendance and the highest level completed. Reading ability was measured by asking respondents to read a sentence, with responses classified as cannot read, able to read parts of a sentence, or able to read the whole sentence. Media exposure was measured by the frequency of reading newspapers or magazines, listening to the radio, and watching television, each categorized as not at all, less than once a week, or at least once a week. Digital access included the frequency of internet use in the past month (not at all, less than once a week, at least once a week, or almost every day), mobile phone ownership (yes/no), and mobile phone use for financial transactions in the past year (yes/no).

#### 2.2.2. Employment

This dimension was assessed using indicators of employment group and type of earnings. Employment was assessed using two questions: whether respondents had worked in the past 12 months and their main type of occupation. Based on the responses, the occupation group was categorized as not working, working in agriculture, or working in non-agricultural sectors. Type of earnings for work was measured by asking whether respondents were paid in cash, in kind, both, or not at all.

#### 2.2.3. Decision-Making

This dimension was assessed through indicators of relative earning and autonomy in financial and household decisions. Relative earnings were assessed by asking respondents whether they earned more than, less than, or about the same as their partner, or whether their partner had no income, and were categorized as ‘whether the respondent earns more than her partner’. Autonomy over income use was measured by asking who usually decides how the respondent’s earnings are used, categorized as independently, jointly, or not involved, and labeled as “decision-making over use of respondent’s income”. Similarly, three household-level decisions—‘decision-making on own health care,’ ‘decision-making on household purchases,’ and ‘decision-making on visiting family’—were assessed and recoded into independently, jointly, or not involved, and labeled accordingly.

### 2.3. Immunization Schedule of Childhood Vaccines and Coverage Measures

The outcome measure of this study was vaccination coverage on schedule among children aged 0 to 2 years, focusing on four key antigens: oral polio vaccine (OPV), DTP, PCV, and measles–rubella (MR). OPV, DTP, and PCV are administered in three doses at 6, 10, and 14 weeks of age, while MR is given in two doses at 9 and 18 months [19]. Cambodia has incorporated 11 of the 13 WHO-recommended routine vaccines for children under 2 years of age, with only rotavirus vaccine and Haemophilus influenzae type b (Hib) not yet included in the national schedule [21]. The detailed schedules for OPV, DTP, PCV, and MR used in this analysis are presented in Appendix A. The multi-dose nature of these vaccines provides a more meaningful measure of women’s empowerment, as completion of multi-dose schedules may reflect a greater degree of sustained maternal decision-making or engagement with health services. For each antigen, children were categorized into “complete”, “incomplete”, and “no vaccination” based on the number of doses received. Coverage was calculated as the proportion of children who had received complete, partial, or no doses among all children who had reached the eligible age for the final dose by 2 years of age. Vaccination coverage was assessed only among children who had reached the recommended age for the final dose of each vaccine.

### 2.4. Demographic and Socioeconomic Factors

Demographic and socioeconomic variables included maternal age (≤19, 20–29, and 30+), residence (urban or rural), wealth (poorest, poorer, middle, richer, and richest), and parity (1, 2, 3+). The CDHS wealth index was constructed using principal component analysis based on household assets, construction materials, and access to water and sanitation facilities, to represent each household’s relative standard of living.

### 2.5. Statistical Analysis

Descriptive analyses were employed to assess trends in women’s empowerment and key socioeconomic characteristics across survey years, and to examine the relationship between women’s empowerment and childhood vaccination coverage using the 2021–22 survey data. Multinomial logistic regression models were constructed to evaluate associations between women’s empowerment and child immunization outcomes after adjusting for maternal demographic and socioeconomic characteristics.

To assess potential multicollinearity among women’s empowerment indicators and covariates, we conducted variance inflation factor (VIF) diagnostics using a linear model including all predictors. All VIF values were below the commonly accepted threshold of 10, indicating no evidence of problematic multicollinearity. Further separate regression models were run in which empowerment variables and covariates were entered individually before being combined in the full model. The directions and magnitudes of the effects were similar across these specifications, suggesting that multicollinearity did not materially affect the estimated associations.

Traditional media variables, such as radio and television, were excluded from regression models because they had declined significantly by 2021. Two-sided *p* values less than 0.05 were considered statistically significant. All analyses were conducted using R version 4.3.3 [22].

## 3. Results

### 3.1. Participants

This study included 3097 women in 2010, 2841 women in 2014, and 3284 women in 2021–22 who had given birth within the preceding 2 years (Table 1).

Between 2010 and 2021–2022, the sociodemographic profile of women changed significantly. The proportion of women aged 30 years or older increased steadily from 28.0% in 2010 to 38.9% in 2021–2022. The urban-rural difference has narrowed, with urban women comprising about one-third by 2021–2022. In 2021–2022, disadvantaged groups formed a larger share, with 28.8% of women in the poorest quintile and 19.6% in the poorer quintile. The share of women with one or three and more children declined, while those with two children rose from 27.0% in 2010 to 36.2% in 2021–2022, becoming the largest group.

Table 2 presents changes in women’s empowerment between 2010 and 2021–2022 across three domains: literacy and information retrieval, employment, and decision-making.

Educational attainment improved steadily, with nearly half of women having completed primary education, while the proportion with no schooling declined to 12.0% in 2021–2022. Reading ability also advanced, as only 1 in 5 women was unable to read a sentence by then. Media exposure, however, showed differential trends. Traditional media use declined markedly: the majority of women neither read newspapers or magazines nor listened to the radio, and two-thirds did not watch television. In contrast, digital access expanded rapidly. By 2021–22, 82.4% of women owned a mobile phone, about half used the internet almost daily, and 21.0% reported engaging in mobile financial transactions.

Occupational patterns shifted considerably between 2010 and 2021–2022. The share of women employed in agriculture declined from 44.5% to 18.1%, while those engaged in non-agricultural work rose from about one-third to nearly half. Regarding the type of earnings, cash compensation increased markedly from 48.7% in 2010 to 82.5% in 2021–2022, whereas in-kind compensation declined significantly from 23.0% to 3.2%.

Regarding earnings relative to their partners, 48.2% of women in 2021–2022 reported earning less, while about one-third contributed a similar amount. Regarding decision-making, 64.3% of women could independently decide how to use their income, although one-third also engaged in joint decision-making in the same survey year. For household-related decisions, joint decision-making predominated: more than half of women made health care decisions with their partners, and the majority shared major purchase and family visit decisions with their partners.

### 3.2. Child Vaccination Coverage by Maternal Sociodemographic Factors, 2021–2022

Regarding the focal vaccines in 2021–2022, childhood immunization coverage was generally high, with completion rates of 79–83% for OPV, DTP, and PCV, and about 53% for measles.

Table 3 and Table 4 present childhood immunization completion rates by maternal sociodemographic characteristics (2021–2022). Regarding maternal age, children of women younger than 19 had substantially lower completion rates than those of older women. Regarding urban-rural differences, rural children were more likely to have incomplete or no vaccination than urban children, with a difference of about 5% across vaccines. By wealth index, children from poorer households consistently had the lowest completion rates for all four vaccines, compared with those from richer and richest households. Regarding parity, children of women with three or more children were more likely to have incomplete vaccination, whereas those of mothers with one or two children had relatively higher completion rates.

### 3.3. Child Vaccination Coverage by Women Empowerment, 2021–2022

Table 5 and Table 6 present childhood immunization completion rates according to domains of women’s empowerment for 2021–2022.

Children whose mothers had no formal education were far less likely to complete vaccinations than those whose mothers had at least secondary education. Among these children, only about two-thirds completed multi-dose vaccines (OPV, DTP, PCV), and barely one-third received measles vaccination. Similarly, while the majority of children of women who could read a full sentence were fully vaccinated, about one-third of children of women who could not read at all were not fully vaccinated. Regarding traditional media use, children of women who did not listen to the radio or watch television had lower vaccination completion rates compared with those whose mothers did. In addition, digital access also revealed substantial disparities: compared to children of mothers who owned a mobile phone, those of non-owners demonstrated lower vaccination completion rates: 71% for OPV, 67% for DTP, and 69% for PCV, versus 46% for measles. Children of women who never accessed the internet and those who did not use mobile phones for financial transactions also had lower vaccination completion rates than children of women who accessed the internet almost daily or had experience with mobile financial transactions.

Compared with women employed in non-agricultural sectors, children of women engaged in agriculture or not employed had lower vaccination completion rates. In terms of type of earnings, compared with other groups, children of women who were only paid in kind had the lowest completion rates: 69% for OPV, 57% for DTP, and 65% for PCV, and only about 31% for measles.

Higher completion rates for OPV, DTP, and PCV (approximately 80%) were observed among children whose mothers participated in household decision-making, compared with those whose mothers did not (approximately 70%). For measles, however, differences across decision-making groups were less pronounced, with completion remaining relatively low in all groups.

### 3.4. Association Between Women Empowerment and Child Vaccination Completion

Table 7 summarizes the adjusted associations between women’s empowerment dimensions and children’s vaccination completion. Overall, most empowerment indicators did not show uniform associations across vaccine groups. Instead, a consistent pattern emerged in which the direction and magnitude of associations for measles vaccination differed from those observed for OPV, DTP, and PCV, even when some estimates did not reach statistical significance. This contrast suggests that empowerment dimensions may influence early multi-dose vaccines and later-scheduled vaccines through different pathways.

Maternal education showed divergent associations across vaccine types. Compared with mothers with secondary or higher education, those with incomplete primary education had higher odds of incomplete OPV and PCV vaccination (adjusted ORs ranging from approximately 1.9 to 2.5), indicating lower completion of early multi-dose vaccines. In contrast, the same group had significantly lower odds of incomplete measles vaccination (OR = 0.29, 95% CI: 0.09–0.94), suggesting a protective association for measles completion. No consistent or statistically significant associations were observed for reading ability, media exposure, internet use, or mobile phone-related indicators across vaccine groups.

Maternal employment status was primarily associated with measles vaccination rather than with OPV, DTP, or PCV. Mothers engaged in agricultural work had more than twice the odds of incomplete measles vaccination compared with non-working mothers (OR = 2.25, 95% CI: 1.08–4.68), whereas employment status showed no statistically significant association with incompletion of early multi-dose vaccines.

Type of earnings further highlighted this contrast. For OPV, DTP, and PCV, children of mothers receiving cash-only earnings were less likely to be incompletely vaccinated (adjusted ORs approximately 0.69–0.72), indicating higher completion of early vaccines compared with children of unpaid mothers. However, for measles vaccination, this association reversed: children of mothers receiving cash earnings had significantly higher odds of incomplete vaccination (OR = 2.60, 95% CI: 1.27–5.32).

Decision-making indicators were significantly associated only with measles vaccination. Compared with women who decided independently, those who jointly decided on the use of their own income had higher odds of incomplete measles vaccination (OR = 2.26, 95% CI: 1.13–4.51). In contrast, joint decision-making regarding a woman’s own health care was associated with lower odds of incomplete measles vaccination (OR = 0.42, 95% CI: 0.21–0.83), indicating higher completion. For OPV, DTP, and PCV, decision-making variables showed similar directions to the reference group but did not reach statistical significance.

Taken together, the regression results indicate that women’s empowerment is positively associated with completion of early multi-dose vaccines, but shows neutral or adverse associations with measles vaccination, particularly for employment- and income-related dimensions. These findings underscore that the policy relevance of empowerment dimensions differs by vaccine schedule and that later-scheduled vaccines such as measles may be more sensitive to constraints affecting sustained service utilization.

## 4. Discussion

### 4.1. Summary of Key Findings

Between 2010 and 2022, women’s empowerment in Cambodia improved markedly across literacy, employment, and decision-making. In 2022, completion of OPV, DTP, and PCV was generally high, whereas measles vaccination lagged, with only about half of children completing the recommended dose. After adjusting for maternal socioeconomic and demographic characteristics, higher maternal education and cash income were associated with higher completion of OPV, DTP, and PCV, but with lower measles coverage, suggesting that the effects of empowerment differed by vaccine type. For measles vaccination, joint decision-making about the use of the respondent’s income was associated with higher odds of incomplete vaccination, whereas joint decision-making about the respondent’s own health care was associated with improved vaccination completion.

### 4.2. Interpretation of Findings

The rise in women’s empowerment in Cambodia, particularly in education and employment, parallels broader gains seen in many LMICs during the late MDG (2000–2015) and early SDG (2015–2030) periods. Under the MDGs, MDG 2 sought universal primary education and the elimination of gender gaps in schooling, while MDG 3 aimed to promote gender equality and empower women. The SDGs then embedded gender more deeply across sectors, primarily through SDG 4 (quality education) and SDG 5 (gender equality). These global initiatives were translated into national actions in Cambodia, such as the Neary Ratanak strategies, which expanded girls’ school enrollment, vocational training opportunities, and women’s participation in non-agricultural employment [23,24,25].

In Cambodia, the national immunization program expanded over the past two decades leading to high overall childhood immunization coverage. With long-standing financial and technical support from the Vaccine Alliance (Gavi), the national immunization programme has strengthened cold-chain capacity, rural outreach, and health-worker training, ensuring a reliable vaccine supply and delivery of routine childhood vaccines at no direct cost to families [18,26]. In some settings, performance-based grants to public health facilities have been used to improve service quality and promote the utilization of essential services, including immunization and maternal and child health services. In addition, cash transfer programmes have provided support for pregnant women and young children [23]. These domestic and external investments have jointly strengthened the capacity of Cambodia’s immunization services delivery, as reflected in OPV, DTP, and PCV coverage rates that exceed those of many other LMICs. In 2022, coverage for all three vaccines was approximately 10 percentage points higher than in the Philippines [27].

The negative associations between certain empowerment indicators and measles vaccination completion may seem counterintuitive but likely reflect differences in vaccine timing and the unique structural and behavioral barriers affecting measles vaccination. Unlike much of the LMIC literature that highlights generally positive links between women’s empowerment and childhood immunization, our antigen-specific findings suggest that the effects of empowerment may be schedule-dependent and can weaken or even reverse for later-scheduled measles vaccinations due to time and service-delivery constraints [1,6]. In this study, we found that women’s empowerment, particularly their work status, was positively associated with the completion of early childhood vaccinations, such as DPT, OPV, and PCV, which are typically scheduled around an infant’s third month. Conversely, women’s work status was negatively associated with completion of measles vaccination, which is generally administered when infants are 9–12 months old. Similarly, studies conducted in Cambodia and other parts of Asia suggest that later-scheduled measles doses often lag behind the early multi-dose infant series, emphasizing the need for sustained engagement in vaccination efforts beyond infancy [28,29,30]. This pattern may reflect a combination of caregiver-related constraints, such as the need to return to work and the cumulative burden of repeated visits, as well as supply-side challenges, including mismatched service hours and limited coverage. These barriers are well-documented contributors to vaccination dropout in LMIC settings [31,32,33]

In Cambodia, the Labour Law entitles women to at least 90 days of paid maternity leave, which can extend to 13–14 weeks in specific sectors. This leave period largely coincides with the recommended timeline for completing OPV, DTP, and PCV vaccinations, allowing mothers to remain at home, care for their newborns, and maintain regular engagement with healthcare providers [34]. This alignment between the immunization schedule and maternity leave policies reduces both the time and economic opportunity costs of early vaccination, thereby contributing to higher vaccination completion rates. In contrast, the measles vaccine is scheduled after many mothers have returned to work. For women in informal or agricultural employment, taking time off to visit health facilities is especially challenging, which may partially explain lower measles vaccination completion rates.

Higher maternal education is positively associated with increased coverage of OPV, DTP, and PCV vaccines. This relationship may be explained by greater health literacy, enhanced autonomy, and the financial benefits of paid work, which together support women’s economic independence and mobility. However, for measles vaccination, we observed the opposite trend: higher levels of education and income, often linked to women’s limited work flexibility, may increase the likelihood that mothers miss or forget the recommended immunization schedule.

The decision-making dimensions were also significantly associated with measles vaccination coverage. Shared decision-making regarding a woman’s health care was associated with a greater likelihood of completing measles vaccination, possibly reflecting a household environment that values health and mutual support, thereby facilitating timely immunization. In contrast, joint decision-making about household income was negatively associated with measles vaccination. This may indicate that households prioritize financial stability over preventive care, potentially limiting mothers’ time and resources for completing later vaccine doses. These findings suggest that the effects of women’s empowerment on immunization are not uniformly positive, vary across domains of empowerment, and are influenced by the broader social and structural context. This complexity is particularly evident for later-scheduled vaccinations such as measles.

### 4.3. Policy and Future Research Implications

This study highlights the critical role of women’s empowerment in impacting childhood vaccination completion. These findings may provide valuable insights for other LMICs experiencing rapid progress in women’s empowerment, emphasizing the importance of targeted strategies to sustain the uptake of later-scheduled vaccines as maternal employment and mobility increase. Given that measles vaccination accounts for roughly 60% of all children’s lives saved through routine immunization [35], prioritizing improvements in measles coverage is of public health importance.

Mitigating the negative association between women’s work status and measles vaccination requires integrating demand-generation strategies, such as health education and tailored reminders, with system-level interventions like extended service hours (e.g., evenings or weekends), proactive defaulter tracking, and reliable outreach to improve follow-up, retention, and uptake of later-scheduled vaccines. Although the results suggest that mobile phone ownership and general internet use are not strongly associated with vaccination completion, evidence from other empirical studies indicates that targeted delivery of health information and reminders through mobile technologies is more influential [36]. Building on this evidence, implementing tailored vaccine campaigns and leveraging digital health strategies, such as SMS reminders, interactive voice messages, or mobile applications, may help inform and remind caregivers about vaccination schedules, particularly in a context where women’s access to mobile phones and the internet is rapidly increasing. Such approaches can improve vaccine coverage by reducing time and access barriers for working mothers.

Further studies could examine the local context and assess the effectiveness of interventions implemented to improve measles coverage in Cambodia. In addition, future research using longitudinal designs, mixed-method approaches, or implementation science could help validate the underlying mechanisms linking women’s empowerment to vaccination uptake considering cultural norms, household dynamics, and local variations in service availability, and identify pathways that cannot be captured through cross-sectional data alone.

### 4.4. Strengths and Limitations

This study has several key strengths. First, it draws on three rounds of nationally representative DHS that employ standardized sampling and quality-control procedures. Second, in the Cambodian context, empirical studies specifically examining women’s empowerment and its association with childhood vaccination remain limited. By analyzing data from three survey waves spanning a 12-year period, this study contributes valuable new evidence to this underexplored area, thereby enriching the existing knowledge base on the relationship between women’s empowerment and immunization outcomes in low- and middle-income countries.

Several limitations of this study should be acknowledged. First, maternal self-reported information, particularly regarding children’s vaccination history, may be subject to recall bias. However, in this study, the primary source of data for children’s vaccination status was vaccination cards, with only about 10% of the sample relying solely on maternal recall. This suggests that the potential impact of recall bias is likely to be minimal. Second, the analysis focused exclusively on women’s empowerment and did not include direct data on program-level interventions, such as economic incentive schemes or outreach activities. As a result, this study cannot disentangle the relative contributions of such interventions and women’s empowerment to achieving higher vaccination coverage. Third, the cross-sectional nature of DHS data prevents establishing causality between women’s empowerment and childhood vaccination outcomes. Thus, the observed associations should be interpreted as correlational. Fourth, several women’s decision-making indicators had missing responses. However, since these missing cases represented less than 5% of the sample, they are unlikely to have a significant impact on the final findings.

## 5. Conclusions

In Cambodia, women have made substantial progress in education, economic participation, and household decision-making. Women’s empowerment is associated with higher completion rates of early multi-dose vaccines (OPV, DTP, and PCV). However, women’s employment status was negatively associated with the completion of later-scheduled measles vaccinations (9–12 months), likely due to accumulated time and access constraints as mothers return to work. To address this issue, strengthening community- and village-level outreach, implementing defaulter tracking and reminder systems within routine child health services, and improving the convenience of vaccination services could help increase measles vaccination coverage.

## Figures and Tables

**Table 1 vaccines-14-00048-t001:** Background characteristics of women who have recently given birth to a child up to 2 years of age before the DHS, Cambodia, 2010–2022.

Characteristics	2010 (N = 3097)	2014 (N = 2841)	2021–2022 (N = 3284)	*p*-Value
**Age**				
19 and less	262 (8.46%)	268 (9.43%)	264 (8.04%)	<0.001
20–29	1969 (63.58%)	1762 (62.02%)	1743 (53.08%)	
30+	866 (27.96%)	811 (28.55%)	1277 (38.89%)	
**Residence ^1^**				
Urban	805 (26.33%)	772 (27.56%)	1075 (33.01%)	<0.001
Rural	2252 (73.67%)	2029 (72.44%)	2182 (66.99%)	
**Wealth**				
Poorest	780 (25.19%)	659 (23.2%)	945 (28.78%)	<0.001
Poorer	581 (18.76%)	528 (18.59%)	642 (19.55%)	
Middle	520 (16.79%)	441 (15.52%)	605 (18.42%)	
Richer	572 (18.47%)	506 (17.81%)	660 (20.1%)	
Richest	644 (20.79%)	707 (24.89%)	432 (13.15%)	
**Parity**				
1	1104 (35.65%)	1163 (40.94%)	1072 (32.64%)	<0.001
2	837 (27.03%)	866 (30.48%)	1190 (36.24%)	
3+	1156 (37.33%)	812 (28.58%)	1022 (31.12%)	

^1^ 40 women in 2010, 40 in 2014, and 27 in 2021 had missing values.

**Table 2 vaccines-14-00048-t002:** Empowerment characteristics of women who have recently given birth to a child up to 2 years of age before the DHS, Cambodia, 2010–2022.

Characteristics	2010 (N = 3097)	2014 (N = 2841)	2021–2022 (N = 3284)	*p*-Value
**Literacy and information retrieval**				
** *Education* **				
No education	608 (19.63%)	360 (12.67%)	393 (11.97%)	<0.001
Incomplete primary	1343 (43.36%)	1060 (37.31%)	975 (29.69%)	
Primary	988 (31.9%)	1147 (40.37%)	1476 (44.95%)	
Secondary and higher	158 (5.1%)	274 (9.64%)	440 (13.4%)	
** *Reading ability ^1^* **				
Cannot read at all	962 (31.06%)	664 (23.38%)	665 (20.25%)	<0.001
Able to read only parts of a sentence	669 (21.6%)	641 (22.57%)	991 (30.18%)	
Able to read a whole sentence	1466 (47.34%)	1534 (54.01%)	1624 (49.45%)	
Other **^2^**	0 (0.00%)	1 (0.04%)	4 (0.12%)	
** *Frequency of reading newspapers or magazines ^3^* **				
Not at all	2107 (68.08%)	2176 (76.59%)	2811 (85.6%)	<0.001
Less than once a week	655 (21.16%)	482 (16.97%)	196 (5.97%)	
At least once a week	333 (10.76%)	183 (6.44%)	277 (8.43%)	
** *Frequency of listening to the radio ^4^* **				
Not at all	1240 (40.05%)	1369 (48.19%)	2920 (88.92%)	<0.001
Less than once a week	906 (29.26%)	745 (26.22%)	186 (5.66%)	
At least once a week	950 (30.68%)	727 (25.59%)	178 (5.42%)	
** *Frequency of watching television ^5^* **				
Not at all	838 (27.08%)	889 (31.29%)	2210 (67.3%)	<0.001
Less than once a week	695 (22.46%)	438 (15.42%)	418 (12.73%)	
At least once a week	1562 (50.47%)	1514 (53.29%)	656 (19.98%)	
** *Frequency of using the internet ^6^* **				
Not at all	-	-	1243 (37.85%)	
Less than once a week	-	-	78 (2.38%)	
At least once a week	-	-	295 (8.98%)	
Almost every day	-	-	1668 (50.79%)	
** *Whether one owns a mobile telephone* **				
No	-	-	578 (17.6%)	
Yes	-	-	2706 (82.4%)	
** *Used a mobile phone for financial transactions (last 12 months)* **				
No	-	-	2593 (78.96%)	
Yes	-	-	691 (21.04%)	
**Employment**				
** *Occupation groups ^7^* **				
Did not work	747 (24.14%)	877 (30.99%)	1062 (33.1%)	<0.001
Agriculture **^8^**	1376 (44.47%)	889 (31.41%)	581 (18.11%)	
Non-agriculture **^9^**	971 (31.38%)	1064 (37.6%)	1565 (48.78%)	
** *Type of earnings for work ^10^* **				
Not paid	60 (2.55%)	92 (4.67%)	150 (6.75%)	<0.001
Cash only	1144 (48.70%)	1709 (86.84%)	1834 (82.54%)	
Cash and in-kind	605 (25.76%)	118 (6.00%)	167 (7.52%)	
In-kind only	540 (22.99%)	49 (2.49%)	71 (3.20%)	
**Decision Making**				
** *Whether the respondent earns more than her partner ^11^* **				
More than him	210 (12.49%)	209 (12.00%)	283 (14.89%)	0.03
Less than him	883 (52.5%)	930 (53.42%)	917 (48.24%)	
About the same	582 (34.6%)	597 (34.29%)	694 (36.51%)	
Husband/partner doesn’t bring in money	7 (0.42%)	5 (0.29%)	7 (0.37%)	
** *Decision-making over the use of the respondent’s income ^12^* **				
Independently	1070 (63.77%)	1155 (66.27%)	1224 (64.29%)	<0.001
Jointly **^13^**	591 (35.22%)	550 (31.55%)	604 (31.72%)	
Not involved **^14^**	17 (1.01%)	38 (2.18%)	76 (3.99%)	
** *Decision-making on own health care ^15^* **				
Independently	1184 (39.66%)	1119 (41%)	1211 (38.46%)	0.4
Jointly **^16^**	1530 (51.26%)	1357 (49.73%)	1640 (52.08%)	
Not involved	271 (9.08%)	253 (9.27%)	298 (9.46%)	
** *Decision-making on household purchases ^17^* **				
Independently	412 (13.83%)	385 (14.11%)	456 (14.49%)	<0.001
Jointly	2395 (80.42%)	2121 (77.75%)	2402 (76.3%)	
Not involved	171 (5.74%)	222 (8.14%)	290 (9.21%)	
** *Decision-making on visiting families ^18^* **				
Independently	590 (19.78%)	537 (19.68%)	558 (17.72%)	<0.001
Jointly	2233 (74.86%)	2082 (76.29%)	2354 (74.75%)	
Not involved	160 (5.36%)	110 (4.03%)	237 (7.53%)	

^1^ 1 woman in 2014 had a missing value. ^2^ Other includes respondents not assessable by the DHS reading test due to “no card with required language” or being blind/visually impaired. ^3^ 2 women in 2010 had a missing value. ^4^ 1 woman in 2010 had a missing value. ^5^ 2 women in 2010 had a missing value. ^6^ 2 women in 2010 had a missing value. ^7^ 3 women in 2010, 11 in 2014, and 76 in 2021 had missing values; the answer “do not know” was also treated as a missing value in this analysis. ^8^ Agriculture includes agricultural self-employed and agricultural employees. ^9^ Non-agriculture includes professional/technical/managerial, clerical, sales, household and domestic, services, skilled manual, and unskilled manual. ^10^ Based on respondents who have worked during the last 12 months, 1 woman in 2010 had a missing value. ^11^ Based on currently married or in union women who were paid cash for their work, 8 women in 2010, 3 in 2014, and 3 in 2021 had missing values; the answer “do not know” was also treated as a missing value in this analysis. ^12^ Based on currently married or in union women who were paid cash for their work, 11 women in 2010, 1 in 2014, and 1 in 2021 had missing values. ^13^ Jointly includes the respondent and the husband/partner, and the respondent and another person. ^14^ Not involved includes husband/partner alone, someone else, and others. ^15^ 112 women in 2010, 112 in 2014, and 135 in 2021 had missing values. ^16^ Jointly includes the respondent and the husband/partner, and the respondent and another person. ^17^ 119 women in 2010, 113 in 2014, and 136 in 2021 had missing values. ^18^ 114 women in 2010, 112 in 2014, and 135 in 2021 had missing values.

**Table 3 vaccines-14-00048-t003:** Association between maternal sociodemographic characteristics and childhood vaccination (OPV and DTP), Cambodia, 2021–2022.

Characteristics	OPV (N = 2786) *n* (%)			DTP (N = 2786) *n* (%)		
	No Vaccination	Incomplete	Complete	No Vaccination	Incomplete	Complete
**Age**						
19 and less	28 (12.33%)	33 (14.54%)	166 (73.13%)	37 (16.3%)	32 (14.1%)	158 (69.6%)
20–29	99 (6.68%)	165 (11.13%)	1219 (82.2%)	157 (10.59%)	165 (11.13%)	1161 (78.29%)
30+	83 (7.71%)	117 (10.87%)	876 (81.41%)	116 (10.78%)	111 (10.32%)	849 (78.9%)
*p* Value	0.02	0.1	0.01	0.02	0.04	0.03
**Residence**						
Urban	58 (6.36%)	85 (9.32%)	769 (84.32%)	84 (9.2%)	84 (9.2%)	744 (81.6%)
Rural	149 (8.02%)	227 (12.22%)	1481 (79.75%)	224 (12.1%)	220 (11.8%)	1413 (76.1%)
*p* Value	0.1	0.03	0.004	0.03	0.04	0.001
**Wealth**						
Poorest	102 (12.72%)	118 (14.71%)	582 (72.57%)	145 (18.08%)	109 (13.59%)	548 (68.33%)
Poorer	25 (4.72%)	63 (11.89%)	442 (83.40%)	45 (8.49%)	64 (12.08%)	421 (79.43%)
Middle	32 (6.13%)	62 (11.88%)	428 (81.99%)	41 (7.85%)	65 (12.45%)	416 (79.69%)
Richer	29 (5.19%)	40 (7.16%)	490 (87.66%)	42 (7.51%)	41 (7.33%)	476 (85.15%)
Richest	22 (5.9%)	32 (8.58%)	319 (85.52%)	37 (9.92%)	29 (7.77%)	307 (82.31%)
*p* Value	<0.001	<0.001	<0.001	<0.001	<0.001	<0.001
**Parity**						
1	62 (6.7%)	96 (10.37%)	768 (82.94%)	103 (11.12%)	92 (9.94%)	731 (78.94%)
2	65 (6.48%)	104 (10.37%)	834 (83.15%)	102 (10.17%)	101 (10.07%)	800 (79.76%)
3+	83 (9.68%)	115 (13.42%)	659 (76.9%)	105 (12.25%)	115 (13.42%)	637 (74.33%)
*p* Value	0.02	0.06	<0.001	0.4	0.03	0.01

**Table 4 vaccines-14-00048-t004:** Association between maternal sociodemographic characteristics and childhood vaccination (PCV and Measles), Cambodia, 2021–2022.

Characteristics	PCV (N = 2786) *n* (%)			Measles (N = 784) *n* (%)		
	No Vaccination	Incomplete	Complete	No Vaccination	Incomplete	Complete
**Age**						
19 and less	36 (15.86%)	32 (14.1%)	159 (70.04%)	11 (20.75%)	10 (18.87%)	32 (60.38%)
20–29	149 (10.05%)	156 (10.52%)	1178 (79.43%)	72 (17.73%)	117 (28.82%)	217 (53.45%)
30+	112 (10.41%)	126 (11.71%)	838 (77.88%)	61 (18.77%)	84 (25.85%)	180 (55.38%)
*p* Value	0.03	0.04	0.04	0.2	0.4	0.6
**Residence**						
Urban	81 (8.9%)	85 (9.3%)	746 (81.8%)	44 (15.9%)	73 (26.4%)	159 (57.6%)
Rural	213 (11.5%)	224 (12.1%)	1420 (76.5%)	100 (19.7%)	138 (27.2%)	270 (53.1%)
*p* Value	0.04	0.04	0.002	0.2	0.9	0.3
**Wealth**						
Poorest	125 (15.59%)	116 (14.46%)	561 (69.95%)	58 (24.89%)	67 (28.76%)	108 (46.35%)
Poorer	44 (8.3%)	62 (11.7%)	424 (80%)	19 (13.19%)	42 (29.17%)	83 (57.64%)
Middle	53 (10.15%)	69 (13.22%)	400 (76.63%)	27 (19.71%)	39 (28.47%)	71 (51.82%)
Richer	47 (8.41%)	36 (6.44%)	476 (85.15%)	25 (16.67%)	40 (26.67%)	85 (56.67%)
Richest	28 (7.51%)	31 (8.31%)	314 (84.18%)	15 (12.5%)	23 (19.17%)	82 (68.33%)
*p* Value	<0.001	<0.001	<0.001	0.01	0.3	0.002
**Parity**						
1	98 (10.58%)	96 (10.37%)	732 (79.05%)	42 (16.34%)	62 (24.12%)	153 (59.53%)
2	90 (8.97%)	107 (10.67%)	806 (80.36%)	40 (14.55%)	80 (29.09%)	155 (56.36%)
3+	109 (12.72%)	111 (12.95%)	637 (74.33%)	62 (24.6%)	69 (27.38%)	121 (48.02%)
*p* Value	0.03	0.2	0.004	0.007	0.4	0.03

**Table 5 vaccines-14-00048-t005:** Empowerment characteristics of women associated with childhood immunization among children aged 14 weeks to 2 years (Polio and DTP), Cambodia, 2021–2022.

Characteristics	Polio (N = 2786) *n* (%)			DTP (N = 2786) *n* (%)		
	No Vaccination	Incomplete	Complete	No Vaccination	Incomplete	Complete
** *Literacy and information retrieval* **						
** *Education* **						
No education	57 (16.47%)	56 (16.18%)	233 (67.34%)	76 (21.97%)	54 (15.61%)	216 (62.43%)
Incomplete primary	61 (7.39%)	121 (14.67%)	643 (77.94%)	97 (11.76%)	108 (13.09%)	620 (75.15%)
Primary	69 (5.55%)	123 (9.89%)	1052 (84.57%)	102 (8.2%)	130 (10.45%)	1012 (81.35%)
Secondary and higher	23 (6.2%)	15 (4.04%)	333 (89.76%)	35 (9.43%)	16 (4.31%)	320 (86.25%)
*p* Value	<0.001	<0.001	<0.001	<0.001	<0.001	<0.001
** *Reading ability* **						
Cannot read at all	77 (13.6%)	97 (17.14%)	392 (69.26%)	107 (18.9%)	92 (16.25%)	367 (64.84%)
Able to read only parts of a sentence	49 (5.74%)	103 (12.06%)	702 (82.2%)	82 (9.6%)	91 (10.66%)	681 (79.74%)
Able to read a whole sentence	83 (6.09%)	113 (8.29%)	1167 (85.62%)	120 (8.8%)	124 (9.1%)	1119 (82.1%)
Other	1 (33.33%)	2 (66.67%)	0 (0.00%)	1 (33.33%)	1 (33.33%)	1 (33.33%)
*p* Value	<0.001	<0.001	<0.001	<0.001	<0.001	<0.001
** *Frequency of reading newspapers or magazines* **						
Not at all	181 (7.6%)	282 (11.83%)	1920 (80.57%)	266 (11.16%)	273 (11.46%)	1844 (77.38%)
Less than once a week	10 (5.85%)	14 (8.19%)	147 (85.96%)	11 (6.43%)	15 (8.77%)	145 (84.8%)
At least once a week	19 (8.19%)	19 (8.19%)	194 (83.62%)	33 (14.22%)	20 (8.62%)	179 (77.16%)
*p* Value	0.7	0.1	0.1	0.05	0.3	0.08
** *Frequency of listening to the radio* **						
Not at all	187 (7.55%)	281 (11.35%)	2008 (81.1%)	274 (11.07%)	276 (11.15%)	1926 (77.79%)
Less than once a week	10 (6.13%)	25 (15.34%)	128 (78.53%)	15 (9.2%)	22 (13.5%)	126 (77.3%)
At least once a week	13 (8.84%)	9 (6.12%)	125 (85.03%)	21 (14.29%)	10 (6.8%)	116 (78.91%)
*p* Value	0.7	0.1	0.1	0.05	0.3	0.08
** *Frequency of watching television* **						
Not at all	157 (8.47%)	222 (11.98%)	1474 (79.55%)	231 (12.47%)	220 (11.87%)	1402 (75.66%)
Less than once a week	23 (6.42%)	37 (10.34%)	298 (83.24%)	27 (7.54%)	40 (11.17%)	291 (81.28%)
At least once a week	30 (5.22%)	56 (9.74%)	489 (85.04%)	52 (9.04%)	48 (8.35%)	475 (82.61%)
*p* Value	0.02	0.3	0.007	0.005	0.06	<0.001
** *Frequency of using the internet* **						
Not at all	108 (10.16%)	145 (13.64%)	810 (76.20%)	150 (14.11%)	138 (12.98%)	775 (72.91%)
Less than once a week	4 (6.15%)	10 (15.38%)	51 (78.46%)	6 (9.23%)	9 (13.85%)	50 (76.92%)
At least once a week	17 (7.20%)	25 (10.59%)	194 (82.20%)	20 (8.47%)	29 (12.29%)	187 (79.24%)
Almost every day	81 (5.70%)	135 (9.49%)	1206 (84.81%)	134 (9.42%)	132 (9.28%)	1156 (81.29%)
*p* Value	<0.001	0.009	<0.001	0.001	0.02	<0.001
** *Whether one owns a mobile telephone* **						
No	70 (13.86%)	76 (15.05%)	359 (71.09%)	94 (18.61%)	71 (14.06%)	340 (67.33%)
Yes	140 (6.14%)	239 (10.48%)	1902 (83.38%)	216 (9.47%)	237 (10.39%)	1828 (80.14%)
*p* Value	<0.001	0.004	<0.001	<0.001	0.02	<0.001
** *Used a mobile phone for financial transactions (last 12 months)* **						
No	169 (7.71%)	268 (12.22%)	1756 (80.07%)	251 (11.45%)	259 (11.81%)	1683 (76.74%)
Yes	41 (6.91%)	47 (7.93%)	505 (85.16%)	59 (9.95%)	49 (8.26%)	485 (81.79%)
*p* Value	0.6	0.004	0.006	0.3	0.02	0.01
**Employment**						
** *Occupation groups* **						
Did not work	74 (8.32%)	106 (11.92%)	709 (79.75%)	98 (11.02%)	111 (12.49%)	680 (76.49%)
Agriculture	49 (9.66%)	69 (13.61%)	389 (76.73%)	73 (14.40%)	64 (12.62%)	370 (72.98%)
Non-agriculture	82 (6.17%)	131 (9.86%)	1116 (83.97%)	129 (9.71%)	126 (9.48%)	1074 (80.81%)
*p* Value	0.02	0.05	<0.001	0.02	0.04	<0.001
** *Type of earnings for work* **						
Not paid	15 (11.45%)	13 (9.92%)	103 (78.63%)	20 (15.27%)	11 (8.40%)	100 (76.34%)
Cash only	93 (5.97%)	167 (10.71%)	1299 (83.32%)	143 (9.17%)	159 (10.20%)	1257 (80.63%)
Cash and in-kind	15 (10.64%)	22 (15.60%)	104 (73.76%)	29 (20.57%)	19 (13.48%)	93 (65.96%)
In-kind only	13 (19.70%)	7 (10.61%)	46 (69.70%)	20 (30.30%)	8 (12.12%)	38 (57.58%)
*p* Value	<0.001	0.3	0.001	<0.001	0.5	<0.001
**Decision Making**						
** *Whether the respondent earns more than her partner* **						
More than him	12 (5.00%)	22 (9.17%)	206 (85.83%)	20 (8.33%)	21 (8.75%)	199 (82.92%)
Less than him	44 (5.72%)	82 (10.66%)	643 (83.62%)	67 (8.71%)	78 (10.14%)	624 (81.14%)
About the same	44 (7.39%)	74 (12.44%)	477 (80.17%)	73 (12.27%)	68 (11.43%)	454 (76.30%)
Husband/partner doesn’t bring in money	0 (0.00%)	1 (16.67%)	5 (83.33%)	0 (0.00%)	1 (16.67%)	5 (83.33%)
*p* Value	0.4	0.5	0.2	0.1	0.6	0.08
** *Decision-making over the use of the respondent’s income* **						
Independently	60 (5.81%)	115 (11.14%)	857 (83.04%)	91 (8.82%)	114 (11.05%)	827 (80.14%)
Jointly	33 (6.47%)	57 (11.18%)	420 (82.35%)	48 (9.41%)	49 (9.61%)	413 (80.98%)
Not involved	7 (10.00%)	8 (11.43%)	55 (78.57%)	21 (30.00%)	6 (8.57%)	43 (61.43%)
*p* Value	0.4	1	0.6	<0.001	0.6	<0.001
** *Decision-making on own health care* **						
Independently	77 (7.54%)	111 (10.87%)	833 (81.59%)	111 (10.87%)	112 (10.97%)	798 (78.16%)
Jointly	98 (7.10%)	159 (11.51%)	1124 (81.39%)	132 (9.56%)	152 (11.01%)	1097 (79.44%)
Not involve	23 (8.78%)	35 (13.36%)	204 (77.86%)	50 (19.08%)	32 (12.21%)	180 (68.70%)
*p* Value	0.6	0.5	0.4	<0.001	0.8	<0.001
** *Decision-making on household purchases* **						
Independently	29 (7.59%)	46 (12.04%)	307 (80.37%)	44 (11.52%)	48 (12.57%)	290 (75.92%)
Jointly	145 (7.11%)	235 (11.53%)	1658 (81.35%)	202 (9.91%)	225 (11.04%)	1611 (79.05%)
Not involve	24 (9.88%)	24 (9.88%)	195 (80.25%)	47 (19.34%)	23 (9.47%)	173 (71.19%)
*p* Value	0.3	0.7	0.8	<0.001	0.5	0.01
** *Decision-making on visiting families* **						
Independently	48 (9.96%)	47 (9.75%)	387 (80.29%)	72 (14.94%)	47 (9.75%)	363 (75.31%)
Jointly	133 (6.73%)	234 (11.84%)	1610 (81.44%)	177 (8.95%)	227 (11.48%)	1573 (79.56%)
Not involve	17 (8.29%)	23 (11.22%)	165 (80.49%)	44 (21.46%)	21 (10.24%)	140 (68.29%)
*p* Value	0.05	0.4	0.8	<0.001	0.5	<0.001

**Table 6 vaccines-14-00048-t006:** Empowerment characteristics of women associated with childhood immunization among children aged 14 weeks to 2 years (PCV) and 18 months to 2 years (Measles), Cambodia, 2021–2022.

Characteristics	PCV (N = 2786) *n* (%)			Measles (N = 784) *n* (%)		
	No Vaccination	Incomplete	Complete	No Vaccination	Incomplete	Complete
**Literacy and information retrieval**						
** *Education* **						
No education	65 (18.79%)	59 (17.05%)	222 (64.16%)	36 (34.95%)	31 (30.1%)	36 (34.95%)
Incomplete primary	92 (11.15%)	119 (14.42%)	614 (74.42%)	39 (17.41%)	67 (29.91%)	118 (52.68%)
Primary	110 (8.84%)	121 (9.73%)	1013 (81.43%)	57 (16.47%)	91 (26.3%)	198 (57.23%)
Secondary and higher	30 (8.09%)	15 (4.04%)	326 (87.87%)	12 (10.81%)	22 (19.82%)	77 (69.37%)
*p* Value	<0.001	<0.001	<0.001	<0.001	0.2	<0.001
** *Reading ability* **						
Cannot read at all	93 (16.43%)	99 (17.49%)	374 (66.08%)	49 (29.52%)	46 (27.71%)	71 (42.77%)
Able to read only parts of a sentence	84 (9.84%)	95 (11.12%)	675 (79.04%)	33 (14.86%)	72 (32.43%)	117 (52.7%)
Able to read a whole sentence	120 (8.8%)	118 (8.66%)	1125 (82.54%)	62 (15.7%)	92 (23.29%)	241 (61.01%)
Other	0 (0.00%)	2 (66.67%)	1 (33.33%)	0 (0.00%)	1 (100.00%)	0 (0.00%)
*p* Value	<0.001	<0.001	<0.001	<0.001	0.03	<0.001
** *Frequency of reading newspapers or magazines* **						
Not at all	257 (10.78%)	280 (11.75%)	1846 (77.47%)	120 (18.07%)	182 (27.41%)	362 (54.52%)
Less than once a week	11 (6.43%)	16 (9.36%)	144 (84.21%)	9 (15.52%)	14 (24.14%)	35 (60.34%)
At least once a week	29 (12.5%)	18 (7.76%)	185 (79.74%)	15 (24.19%)	15 (24.19%)	32 (51.61%)
*p* Value	0.5	0.1	0.2	0.4	0.8	0.6
** *Frequency of listening to the radio* **						
Not at all	262 (10.58%)	282 (11.39%)	1932 (78.03%)	125 (17.83%)	190 (27.1%)	386 (55.06%)
Less than once a week	14 (8.59%)	25 (15.34%)	124 (76.07%)	9 (21.43%)	12 (28.57%)	21 (50%)
At least once a week	21 (14.29%)	7 (4.76%)	119 (80.95%)	10 (24.39%)	9 (21.95%)	22 (53.66%)
*p* Value	0.5	0.1	0.2	0.4	0.8	0.6
** *Frequency of watching television* **						
Not at all	219 (11.82%)	220 (11.87%)	1414 (76.31%)	100 (19.27%)	151 (29.09%)	268 (51.64%)
Less than once a week	28 (7.82%)	47 (13.13%)	283 (79.05%)	21 (20.59%)	27 (26.47%)	54 (52.94%)
At least once a week	50 (8.7%)	47 (8.17%)	478 (83.13%)	23 (14.11%)	33 (20.25%)	107 (65.64%)
*p* Value	0.02	0.02	0.002	0.3	0.08	0.007
** *Frequency of using the internet* **						
Not at all	138 (12.98%)	141 (13.26%)	784 (73.75%)	63 (21.14%)	73 (24.50%)	162 (54.36%)
Less than once a week	5 (7.69%)	10 (15.38%)	50 (76.92%)	4 (16.00%)	8 (32.00%)	13 (52.00%)
At least once a week	23 (9.75%)	33 (13.98%)	180 (76.27%)	12 (18.46%)	21 (32.31%)	32 (49.23%)
Almost every day	131 (9.21%)	130 (9.14%)	1161 (81.65%)	65 (16.41%)	109 (27.53%)	222 (56.06%)
*p* Value	0.004	0.02	<0.001	0.5	0.5	0.8
** *Whether one owns a mobile telephone* **						
No	81 (16.04%)	75 (14.85%)	349 (69.11%)	38 (27.94%)	35 (25.74%)	63 (46.32%)
Yes	216 (9.47%)	239 (10.48%)	1826 (80.05%)	106 (16.36%)	176 (27.16%)	366 (56.48%)
*p* Value	<0.001	0.006	<0.001	0.002	0.8	0.04
** *Used a mobile phone for financial transactions (last 12 months)* **						
No	241 (10.99%)	261 (11.90%)	1691 (77.11%)	125 (20.36%)	167 (27.20%)	322 (52.44%)
Yes	56 (9.44%)	53 (8.94%)	484 (81.62%)	19 (11.18%)	44 (25.88%)	107 (62.94%)
*p* Value	0.3	0.05	0.02	0.009	0.8	0.02
**Employment**						
** *Occupation groups* **						
Did not work	103 (11.59%)	99 (11.14%)	687 (77.28%)	42 (19.72%)	47 (22.07%)	124 (58.22%)
Agriculture	56 (11.05%)	77 (15.19%)	374 (73.77%)	36 (22.93%)	45 (28.66%)	76 (48.41%)
Non-agriculture	130 (9.78%)	130 (9.78%)	1069 (80.44%)	64 (16.08%)	113 (28.39%)	221 (55.53%)
*p* Value	0.005	0.5	0.006	0.2	0.2	0.2
** *Type of earnings for work* **						
Not paid	19 (14.50%)	14 (10.69%)	98 (74.81%)	7 (21.21%)	9 (27.27%)	17 (51.52%)
Cash only	141 (9.04%)	172 (11.03%)	1246 (79.92%)	76 (15.97%)	142 (29.83%)	258 (54.20%)
Cash and in-kind	19 (13.48%)	21 (14.89%)	101 (71.63%)	13 (28.26%)	8 (17.39%)	25 (54.35%)
In-kind only	15 (22.73%)	8 (12.12%)	43 (65.15%)	6 (37.50%)	5 (31.25%)	5 (31.25%)
*p* Value	0.3	0.02	0.01	0.03	0.4	0.3
**Decision Making**						
** *Whether the respondent earns more than her partner* **						
More than him	26 (10.83%)	20 (8.33%)	194 (80.83%)	10 (14.08%)	23 (32.39%)	38 (53.52%)
Less than him	57 (7.41%)	92 (11.96%)	620 (80.62%)	29 (13.36%)	70 (32.26%)	118 (54.38%)
About the same	62 (10.42%)	73 (12.27%)	460 (77.31%)	42 (21.32%)	46 (23.35%)	109 (55.33%)
Husband/partner doesn’t bring in money	1 (16.67%)	0 (0.00%)	5 (83.33%)	0 (0.00%)	0 (0.00%)	1 (100.00%)
*p* Value	0.5	0.5	0.2	0.2	0.2	0.8
** *Decision-making over the use of the respondent’s income* **						
Independently	87 (8.43%)	125 (12.11%)	820 (79.46%)	52 (16.20%)	91 (28.35%)	178 (55.45%)
Jointly	50 (9.80%)	50 (9.80%)	410 (80.39%)	25 (16.89%)	47 (31.76%)	76 (51.35%)
Not involved	9 (12.86%)	11 (15.71%)	50 (71.43%)	4 (23.53%)	1 (5.88%)	12 (70.59%)
*p* Value	0.4	0.2	0.2	0.7	0.08	0.3
** *Decision-making on own health care* **						
Independently	106 (10.38%)	105 (10.28%)	810 (79.33%)	53 (18.03%)	76 (25.85%)	165 (56.12%)
Jointly	142 (10.28%)	156 (11.30%)	1083 (78.42%)	71 (19.24%)	103 (27.91%)	195 (52.85%)
Not involve	30 (11.45%)	44 (16.79%)	188 (71.76%)	12 (16.44%)	18 (24.66%)	43 (58.90%)
*p* Value	0.8	0.01	0.03	0.8	0.8	0.5
** *Decision-making on household purchases* **						
Independently	40 (10.47%)	46 (12.04%)	296 (77.49%)	18 (17.82%)	23 (22.77%)	60 (59.41%)
Jointly	203 (9.96%)	229 (11.24%)	1606 (78.80%)	108 (18.91%)	164 (28.72%)	299 (52.36%)
Not involve	35 (14.40%)	30 (12.35%)	178 (73.25%)	10 (15.62%)	10 (15.62%)	44 (68.75%)
*p* Value	0.1	0.8	0.1	0.8	0.05	0.03
** *Decision-making on visiting families* **						
Independently	62 (12.86%)	46 (9.54%)	374 (77.59%)	23 (16.91%)	32 (23.53%)	81 (59.56%)
Jointly	189 (9.56%)	228 (11.53%)	1560 (78.91%)	103 (18.93%)	154 (28.31%)	287 (52.76%)
Not involve	26 (12.68%)	31 (15.12%)	148 (72.20%)	10 (17.86%)	11 (19.64%)	35 (62.50%)
*p* Value	0.06	0.1	0.08	0.9	0.2	0.2

**Table 7 vaccines-14-00048-t007:** Factors associated with incomplete status vaccination among children in Cambodia, based on multinomial logistic regression analysis (2021–2022).

	OPV (N = 2786)/n (%)		DTP (N = 2786)/n (%)		PCV (N = 2786)/n (%)		Measles (N = 784)/n (%)	
	Adjusted OR (95% CI)	*p*-Value	Adjusted OR (95% CI)	*p*-Value	Adjusted OR (95% CI)	*p*-Value	Adjusted OR (95% CI)	*p*-Value
**Literacy and information retrieval**								
** *Education* **								
Secondary and higher	Ref.	Ref.	Ref.	Ref.	Ref.	Ref.	Ref.	Ref.
No education	2.06 (0.89, 4.79)	0.1	2.08 (0.94, 4.61)	0.1	2.76 (1.25, 6.08)	0.01	0.19 (0.03, 1.29)	0.09
Incomplete primary	1.94 (1.04, 3.63)	0.04	1.70 (0.95, 3.04)	0.07	2.49 (1.39, 4.46)	0.002	0.29 (0.09, 0.94)	0.04
Primary	1.25 (0.76, 2.06)	0.4	1.15 (0.73, 1.80)	0.6	1.46 (0.92, 2.32)	0.1	0.53 (0.22, 1.29)	0.2
** *Reading ability* **								
Able to read a whole sentence	Ref.	Ref.	Ref.	Ref.	Ref.	Ref.	Ref.	Ref.
Cannot read at all	1.12 (0.62, 2.03)	0.7	1.13 (0.64, 2.00)	0.7	0.97 (0.55, 1.69)	0.9	3.09 (0.78, 12.16)	0.1
Able to read only parts of a sentence	0.93 (0.62, 1.40)	0.7	0.90 (0.61, 1.32)	0.6	0.87 (0.60, 1.27)	0.5	1.07 (0.51, 2.23)	0.9
** *Frequency of reading newspapers or magazines* **								
At least once a week	Ref.	Ref.	Ref.	Ref.	Ref.	Ref.	Ref.	Ref.
Less than once a week	0.77 (0.37, 1.60)	0.5	0.66 (0.33, 1.33)	0.3	0.71 (0.36, 1.42)	0.3	2.05 (0.57, 7.34)	0.3
Not at all	0.80 (0.49, 1.32)	0.4	0.77 (0.49, 1.21)	0.3	0.80 (0.51, 1.27)	0.3	1.78 (0.74, 4.27)	0.2
** *Frequency of using the internet* **								
Almost every day	Ref.	Ref.	Ref.	Ref.	Ref.	Ref.	Ref.	Ref.
At least once a week	1.12 (0.68, 1.85)	0.7	1.04 (0.65, 1.67)	0.9	1.19 (0.75, 1.87)	0.5	1.23 (0.38, 3.95)	0.7
Less than once a week	1.90 (0.88, 4.11)	0.1	1.48 (0.68, 3.21)	0.3	1.80 (0.85, 3.80)	0.1	0.88 (0.16, 4.94)	0.9
Not at all	1.35 (0.94, 1.96)	0.1	1.20 (0.85, 1.71)	0.3	1.37 (0.97, 1.93)	0.07	1.14 (0.56, 2.33)	0.7
** *Whether one owns a mobile telephone* **								
Yes	Ref.	Ref.	Ref.	Ref.	Ref.	Ref.	Ref.	Ref.
No	1.27 (0.84, 1.91)	0.3	1.22 (0.82, 1.83)	0.3	1.14 (0.77, 1.69)	0.5	1.03 (0.40, 2.68)	0.9
** *Used a mobile phone for financial transactions (last 12 months)* **								
Yes	Ref.	Ref.	Ref.	Ref.	Ref.	Ref.	Ref.	Ref.
No	0.91 (0.63, 1.31)	0.6	0.91 (0.65, 1.28)	0.6	0.89 (0.64, 1.25)	0.5	1.03 (0.53, 2.02)	0.9
**Employment**								
** *Occupation groups* **								
Did not work	Ref.	Ref.	Ref.	Ref.	Ref.	Ref.	Ref.	Ref.
Agriculture	0.76 (0.55, 1.06)	0.1	0.80 (0.59, 1.09)	0.16	0.80 (0.59, 1.08)	0.1	2.25 (1.08, 4.68)	0.03
Non-agriculture	0.87 (0.64, 1.19)	0.4	1.06 (0.79, 1.43)	0.7	0.84 (0.63, 1.13)	0.3	1.62 (0.80, 3.29)	0.2
** *Type of earnings for work* **								
Not paid	Ref.	Ref.	Ref.	Ref.	Ref.	Ref.	Ref.	Ref.
Cash only	0.69 (0.50, 0.95)	0.0	0.69 (0.51, 0.94)	0.02	0.72 (0.53, 0.98)	0.04	2.60 (1.27, 5.32)	0.01
Cash and in-kind	0.96 (0.66, 1.39)	0.8	1.22 (0.86, 1.74)	0.3	0.93 (0.66, 1.33)	0.7	1.41 (0.61, 3.25)	0.4
In-kind only	/	/	/	/	/	/	/	/
**Decision Making**								
** *Whether the respondent earns more than her partner* **								
More than him	Ref.	Ref.	Ref.	Ref.	Ref.	Ref.	Ref.	Ref.
Less than him	0.90 (0.58, 1.40)	0.7	0.90 (0.60, 1.35)	0.6	0.82 (0.55, 1.22)	0.3	1.55 (0.71, 3.39)	0.3
About the same	1.38 (0.89, 2.16)	0.2	1.51 (0.99, 2.29)	0.05	1.19 (0.79, 1.78)	0.4	1.09 (0.51, 2.36)	0.8
Husband/partner doesn’t bring in money	0.96 (0.10, 9.48)	1.0	0.97 (0.10, 9.41)	1.0	0.69 (0.07, 6.70)	0.8	/	/
** *Decision-making over the use of the respondent’s income* **								
Independently	Ref.	Ref.	Ref.	Ref.	Ref.	Ref.	Ref.	Ref.
Jointly	1.09 (0.79, 1.51)	0.6	1.01 (0.74, 1.38)	1.0	0.98 (0.72, 1.33)	0.9	2.26 (1.13, 4.51)	0.02
Not involved	1.02 (0.47, 2.21)	1.0	1.50 (0.76, 2.96)	0.2	0.84 (0.42, 1.69)	0.6	/	/
** *Decision-making on own health care* **								
Independently	Ref.	Ref.	Ref.	Ref.	Ref.	Ref.	Ref.	Ref.
Jointly	1.01 (0.72, 1.43)	1.0	0.97 (0.70, 1.34)	0.8	1.10 (0.80, 1.52)	0.6	0.42 (0.21, 0.83)	0.01
Not involve	1.33 (0.78, 2.27)	0.3	1.51 (0.92, 2.47)	0.1	1.35 (0.82, 2.21)	0.2	1.24 (0.32, 4.87)	0.8
** *Decision-making on household purchases* **								
Independently	Ref.	Ref.	Ref.	Ref.	Ref.	Ref.	Ref.	Ref.
Jointly	0.96 (0.61, 1.50)	0.9	0.97 (0.64, 1.48)	0.9	0.84 (0.56, 1.28)	0.4	0.98 (0.39, 2.45)	1.0
Not involve	1.11 (0.50, 2.42)	0.8	0.93 (0.44, 1.94)	0.8	1.25 (0.61, 2.55)	0.5	0.30 (0.05, 1.70)	0.2
** *Decision-making on visiting families* **								
Independently	Ref.	Ref.	Ref.	Ref.	Ref.	Ref.	Ref.	Ref.
Jointly	0.70 (0.47, 1.03)	0.1	0.71 (0.49, 1.03)	0.07	0.78 (0.54, 1.13)	0.2	1.20 (0.53, 2.74)	0.7
Not involve	0.68 (0.31, 1.49)	0.3	1.14 (0.55, 2.34)	0.7	1.06 (0.52, 2.16)	0.9	1.52 (0.26, 8.81)	0.6

## Data Availability

All data used in this study are publicly available from the DHS Program database at https://www.dhsprogram.com/data/Access-Instructions.cfm (accessed on 15 August 2024).

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
