# Peer review of "Trends in Women’s Empowerment and Their Association with Childhood Vaccination in Cambodia: Evidence from Demographic and Health Surveys (2010–2022)"

_vaccines, 2025, doi:10.3390/vaccines14010048_

Round 1
Reviewer 1 Report
Comments and Suggestions for Authors
This is an interesting study assessing the impact of changes in women's statuses in Cambodia with children vaccination trend. A few issues need to be addressed before this manuscript can be accepted.
- Add "Cambodia" and "public health" to the keywords list
- Line 112: Please cite the reference for the CDHS technical documentation.
- Materials and methods: Please provide the Cambodian child immunization schedule in the supplementary materials, so that the reader can follow visually the vaccines that are mandated during the first two years of life. Make sure to cite it in the text.
- Statistical analysis: Please indicate the cutoff P value that indicated statistical significance.
- Table 2: The footnotes corresponding to number 1 and 2 are unnecessary as the items are clear. Regarding the footnote no. 4 , what do you mean y "no card with required language"? Please clarify.
- Discussion: The paragraph on lines 342-353 should eb moved to the introduction.
- Lines 363-368: Please reword these statements to make them sound like potential reasons for higher vaccination rates except for measles as it currently sounds like a concrete evidence. This is because the survey didn't specifically address maternity leave and the fact that many of the included women were either unemployed or self-employed.
- It was kind of surprising that owning a mobile phone or internet exposure wasn't associated with increased vaccination rates since people sometimes may get exposed to such info on the web. The discussion lacks a comment on this aspect. Please mention it between the education and decision-making paragraphs.
- Limitations: One important limitation is the numerous missing values reported under Table 2, which may have impacted the findings reported in subsequent tables.
- It is worth mentioning in the introduction or the discussion if the Cambodian MoH implements vaccination programs in schools (or daycares if they're used by working women), public places, or through home visits. If not, this could be mentioned as a recommendation under "Policy and Future Research Implications" section.
Author Response
- Add "Cambodia" and "public health" to the keywords list
Thank you for this suggestion. We have added both terms to the keywords list (page 1, lines 43–44).
- Line 112: Please cite the reference for the CDHS technical documentation.
Thank you for your comment. The CDHS technical documentation is already cited as Reference 19 in our manuscript. This reference includes the full description of the sampling frame and methodology. As outlined on page 37 of the CDHS 2021–22 Final Report, the survey used the 2019 General Population Census as the sampling frame, adopted a two-stage stratified sampling design (selection of 709 clusters and 30 households per cluster), and included all eligible women aged 15–49. We have ensured that this citation is clearly indicated in the text.
- Materials and methods: Please provide the Cambodian child immunization schedule in the supplementary materials, so that the reader can follow visually the vaccines that are mandated during the first two years of life. Make sure to cite it in the text.
Thank you for this suggestion. We have made the following revisions:
Added an explanation on page 4, lines 162-166, summarizing how many WHO-recommended vaccines are included or not included in Cambodia's national schedule.
Added Supplementary Table S1, which presents the schedule of vaccines relevant to this study, based on WHO documentation (Reference 28).
We want to clarify that our analysis is limited to vaccines collected in DHS surveys. Cambodia's National Immunization Program is not mandatory, but vaccines are provided free of charge at public health facilities. These additions provide readers with a clearer understanding of the national immunization context and the specific vaccines captured by DHS.
- Statistical analysis: Please indicate the cutoff P value that indicated statistical significance.
We appreciate this comment. We have now specified the statistical significance threshold in Section 2.5 and added the sentence: “Two-sided p values less than 0.05 were considered statistically significant.” (page 5, lines 197–199).
- Table 2: The footnotes corresponding to number 1 and 2 are unnecessary as the items are clear. Regarding the footnote no. 4 , what do you mean y "no card with required language"? Please clarify.
Thank you for your thoughtful comment. Footnotes 1 and 2 have been removed as suggested.
Regarding footnote 4, we note that both “no card with required language” and “blind/visually impaired” are original DHS response categories for the reading ability variable (V155). “No card with required language” indicates that the DHS reading test could not be administered because the sentence card was not available in a language the respondent understood. This clarification has been added to Table 2 in the revised manuscript.
- Discussion: The paragraph on lines 342-353 should eb moved to the introduction.
We appreciate this suggestion. However, this paragraph provides contextual interpretation specific to our findings on trends in women’s empowerment, one of the primary aims of this study, rather than general background information. We therefore retained it in the Discussion section to support interpretation of the results.
- Lines 363-368: Please reword these statements to make them sound like potential reasons for higher vaccination rates except for measles as it currently sounds like a concrete evidence. This is because the survey didn't specifically address maternity leave and the fact that many of the included women were either unemployed or self-employed.
Thank you for this important point. These interpretations were developed in consultation with Cambodian partners and are now explicitly presented as plausible contextual factors rather than empirically measured mechanisms.
- It was kind of surprising that owning a mobile phone or internet exposure wasn't associated with increased vaccination rates since people sometimes may get exposed to such info on the web. The discussion lacks a comment on this aspect. Please mention it between the education and decision-making paragraphs.
Thank you for this insightful comment. We have noted that although mobile phone ownership was not associated with vaccination completion in our study, evidence from other settings shows that ownership alone often does not influence health behaviors. Instead, mobile-based delivery of targeted health information, such as SMS reminders or mHealth messaging, has demonstrated positive impacts on vaccination uptake. This point has been incorporated into the Policy and Future Research Implications section (page 18, lines 422–429).
- Limitations: One important limitation is the numerous missing values reported under Table 2, which may have impacted the findings reported in subsequent tables.
Thank you for raising this issue. Although some decision-making variables had missing values, these accounted for less than 5% of the sample and were concentrated within a small subset of indicators, making substantial impact on the regression estimates unlikely. Nonetheless, we acknowledge this as a limitation and have added it as the fourth point in the Limitations section (page 19, lines 468-470).
- It is worth mentioning in the introduction or the discussion if the Cambodian MoH implements vaccination programs in schools (or daycares if they're used by working women), public places, or through home visits. If not, this could be mentioned as a recommendation under "Policy and Future Research Implications" section.
Thank you for this valuable recommendation. Routine childhood vaccinations in Cambodia are primarily delivered through public health facilities rather than workplaces, schools, or home visits. We have incorporated this point into the Discussion (page 17, lines 373–377), noting that incentives and outreach strategies delivered through public facilities may help reduce access barriers for working mothers.
Reviewer 2 Report
Comments and Suggestions for Authors
The paper is on women’s empowerment and childhood vaccination in Cambodia, using DHS data (2010–2022). Addressing the following items will improve the paper:
-The narrative is rich but too descriptive at times, sharpening the causal interpretation, especially for the opposite direction of associations in measles versus OPV/DTP/PCV,would strengthen the mechanistic coherence and avoid over-attributing structural causes without supportive data.
-The regression framework needs clearer justification regarding model selection, handling of multicollinearity among empowerment variables, and the rationale for including certain covariates while excluding others,this will make the analytical pathway more rigorous and reproducible.
-The discussion introduces policy explanations (maternity leave alignment, informal labor time-constraints, household economics) that are plausible but untested in this dataset; explicitly separating evidence-based findings from contextual hypotheses will add conceptual precision.
-The manuscript would benefit from a stronger limitations section on temporality and residual confounding, and a more forward-looking path describing how future mixed-methods or longitudinal designs could validate the complex empowerment–vaccination mechanisms observed here.
Author Response
The paper is on women’s empowerment and childhood vaccination in Cambodia, using DHS data (2010–2022). Addressing the following items will improve the paper:
-The narrative is rich but too descriptive at times, sharpening the causal interpretation, especially for the opposite direction of associations in measles versus OPV/DTP/PCV,would strengthen the mechanistic coherence and avoid over-attributing structural causes without supportive data.
Thank you for this important comment. We agree that the cross-sectional nature of the DHS data limits our ability to make strong causal claims. In response, we have revised the manuscript to sharpen and temper the causal interpretation of our findings. Specifically, in the Strengths and Limitations section, we now explicitly acknowledge that temporality cannot be established, and we emphasize that all associations, including the contrasting patterns observed for measles versus OPV/DTP/PCV, should be interpreted as correlational rather than strictly causal (page 19, lines 465–467).
-The regression framework needs clearer justification regarding model selection, handling of multicollinearity among empowerment variables, and the rationale for including certain covariates while excluding others,this will make the analytical pathway more rigorous and reproducible.
Thank you for this important comment. We have strengthened the methodological justification in the revised manuscript. First, because the outcome contains three unordered categories (complete, incomplete, and no vaccination), we considered that multinomial logistic regression was selected as the appropriate model for estimating these associations.
Second, regarding multicollinearity among women’s empowerment variables, we conducted variance inflation factor (VIF) diagnostics and examined correlations across empowerment dimensions. All VIF values were below the commonly accepted threshold of 10, indicating no evidence of problematic multicollinearity. We also conducted sensitivity analyses by estimating regression models in which empowerment indicators and covariates were entered separately before being combined in the full model. The directions and magnitudes of the estimated effects were similar across these specifications, further confirming that multicollinearity did not materially affect the findings. A concise description of these procedures has been added to the Statistical Analysis section (page 5, lines 188–195).
Finally, covariate selection was guided by prior literature and by theoretical relevance to childhood immunization in low- and middle-income settings, and this rationale is now explicitly stated in the Methods to enhance transparency and reproducibility.
-The discussion introduces policy explanations (maternity leave alignment, informal labor time-constraints, household economics) that are plausible but untested in this dataset; explicitly separating evidence-based findings from contextual hypotheses will add conceptual precision.
We appreciate this thoughtful observation. We agree that the policy explanations discussed, such as the alignment between maternity leave and early-dose vaccination schedules, and time and opportunity costs faced by women in informal or agricultural employment, cannot be directly tested with the current DHS data. These interpretations were developed in consultation with Cambodian partners and are intended as context-informed hypotheses rather than empirically verified mechanisms.
-The manuscript would benefit from a stronger limitations section on temporality and residual confounding, and a more forward-looking path describing how future mixed-methods or longitudinal designs could validate the complex empowerment–vaccination mechanisms observed here.
Thank you for this insightful suggestion. We have strengthened the Strengths and Limitations section by explicitly acknowledging the constraints of temporality inherent in cross-sectional DHS data and clarifying that the associations observed, including the contrasting patterns for measles versus OPV/DTP/PCV, should be interpreted as correlational rather than causal. We also expanded the future research on confounding by noting that residual confounding from unmeasured factors, such as cultural norms, household dynamics, and local variation in service availability, may influence both women’s empowerment and vaccination behaviors (page 18, lines 435–436).
In addition, we revised the Policy and Future Research Implications section to outline a more forward-looking research agenda. We now highlight the value of longitudinal, mixed-methods, and implementation-science designs for validating the mechanisms linking women’s empowerment to childhood immunization and for identifying pathways that cannot be captured through cross-sectional surveys alone (page 18, lines 433–438). These additions, we hope, provide a clearer direction for future research and enhance the manuscript’s conceptual contribution.
Reviewer 3 Report
Comments and Suggestions for Authors
The manuscript studies long-term trends in women’s empowerment in Cambodia and analyzes how different empowerment dimensions are associated with childhood vaccination completion using nationally representative DHS data from 2010–2022. it is important because it moves beyond aggregate vaccination coverage to explore how social and gender dynamics influence vaccine uptake, highlighting vaccine-specific patterns specially for measles, that have direct implications for immunization policy and gender-responsive health interventions in LMIC countries.
Before publication the following minor changes should be done
- In the Abstract the negative or inverse associations observed for measles are central to the paper’s novelty and should be highlighted more clearly as a key finding rather than appearing as a secondary observation.
- The abstract should more explicit states the study’s main conceptual contribution, the differential association between empowerment and measles vaccination compared with other vaccines.
- The introduction should be strengthened by a clearer articulation of the theoretical pathways linking empowerment dimensions to multi-dose versus later-schedule vaccines such as measles. The current framing implicitly assumes uniform mechanisms across vaccines, yet the results show otherwise; Authors should acknowledging this possibility upfront .
- the authors Should more explicitly state how findings may (or may not) generalize to similar LMIC contexts.
- In my opinion, the bibliography relies on a relatively limited number of peer-reviewed scientific articles, with a substantial proportion of references consisting of reports, databases, and institutional documents. While these sources are useful for contextual and descriptive purposes, the manuscript would benefit from incorporating more empirical journal articles, particularly those analyzing trends in childhood vaccination coverage for specific antigens such as measles and other routine vaccines in Cambodia and the broader Asian region. Expanding the scientific literature base would help better situate the findings within existing regional evidence, strengthen the analytical depth, and enhance the manuscript’s contribution to the academic debate on vaccination dynamics and women’s empowerment in Asia.
- Empowerment is treated largely as a collection of individual indicators rather than as composite or latent constructs, which raises concerns about conceptual fragmentation and multiple testing.
- The rationale for excluding traditional media variables from regression models, despite presenting them descriptively, should be better justified analytically, because this decision may affect comparability with prior studies.
- The cross-sectional nature of the regression analysis using only 2021–22 data limits causal interpretation, which should be more explicitly acknowledged in the material and methods and in the discussion.
- The large number of stratified tables makes it difficult to identify the most policy-relevant findings. The regression results, especially the contrasting associations for measles versus OPV/DTP/PCV, are scientifically interesting but the authors should include a clearer synthesis in the text, rather than presented them primarily as statistical outputs.
- Some associations appear counterintuitive (eg higher empowerment linked to lower measles completion), which calls for stronger interpretive guidance in the discussion.
- The discussion is somewhat speculative regarding the underlying mechanisms. The manuscript should have a deeper engagement with health system factors such as service delivery timing, caregiver fatigue, or missed opportunities at later child ages, which could interact with empowerment in complex ways.
- As mention before .the discussion should incorporate more academic papers . The reference list relies heavily on reports and databases. The discussion would benefit from adding a small number of recent peer-reviewed studies on antigen-specific vaccination trends (Measless, DTP, ect) particularly measles, in Cambodia or the Asian region.
- The authors needs to better situate the findings within the broader gender and immunization literature .the discussion should incorporate more academic papers by contrasting them more explicitly with studies that report uniformly positive empowerment effects.
- The conclusions should be sharpened by more clearly distinguishing between empowerment as a facilitator of early-life vaccination versus a potentially insufficient factor for later-schedule vaccines like measles.
- The policy implications Should be stronger, the authors should explicitly recommended integrated strategies that combine women’s empowerment with system-level interventions targeting follow-up and retention in immunization programs.
Author Response
- In the Abstract the negative or inverse associations observed for measles are central to the paper’s novelty and should be highlighted more clearly as a key finding rather than appearing as a secondary observation.
Thank you for this important suggestion. We revised the Abstract (Results) to foreground the vaccine-specific, differential association for measles compared with other vaccines (Page 1, Lines 33–37).
- The abstract should more explicit states the study’s main conceptual contribution, the differential association between empowerment and measles vaccination compared with other vaccines.
We very appreciate this helpful recommendation. We further strengthened the Abstract by (i) adding that employment status showed its main association with measles completion rather than early multi-dose vaccines (Page 1, Lines 32–33) and (ii) revising the Abstract (Conclusions) to clarify that later-scheduled vaccines may face additional structural and behavioral barriers as children grow older, and to specify actionable policy implications focused on follow-up and retention for later-dose vaccines (Page 1–2, Lines 39–43).
- The introduction should be strengthened by a clearer articulation of the theoretical pathways linking empowerment dimensions to multi-dose versus later-schedule vaccines such as measles. The current framing implicitly assumes uniform mechanisms across vaccines, yet the results show otherwise; Authors should acknowledging this possibility upfront .
Thank you for this valuable point. We added two sentences to the Introduction to explicitly acknowledge schedule-dependent pathways and to distinguish early multi-dose completion from dropout to later-scheduled vaccines such as measles (Page 2, Lines 66–74).
- the authors Should more explicitly state how findings may (or may not) generalize to similar LMIC contexts.
Thank you for raising this issue. We added an explicit statement in the Policy and Future Research Implications section noting that our findings may offer lessons for other LMICs experiencing rapid gains in women’s empowerment, particularly regarding sustaining uptake of later-scheduled vaccines as maternal employment and mobility increase (Page 19, Lines 459–462).
- In my opinion, the bibliography relies on a relatively limited number of peer-reviewed scientific articles, with a substantial proportion of references consisting of reports, databases, and institutional documents. While these sources are useful for contextual and descriptive purposes, the manuscript would benefit from incorporating more empirical journal articles, particularly those analyzing trends in childhood vaccination coverage for specific antigens such as measles and other routine vaccines in Cambodia and the broader Asian region. Expanding the scientific literature base would help better situate the findings within existing regional evidence, strengthen the analytical depth, and enhance the manuscript’s contribution to the academic debate on vaccination dynamics and women’s empowerment in Asia.
We very appreciate this constructive suggestion. We expanded the peer-reviewed literature base, adding empirical studies on antigen-specific vaccination trends (especially measles) in Cambodia and the Asian region, and additional peer-reviewed evidence to support system-level mechanisms (see also response to Comment 12). We also added Ntenda et al. (2022) on determinants of pentavalent and measles dropout.
- Empowerment is treated largely as a collection of individual indicators rather than as composite or latent constructs, which raises concerns about conceptual fragmentation and multiple testing.
Thank you for highlighting this important methodological concern. We clarified the analytic rationale in the Methods section, stating that empowerment indicators were analyzed separately because different dimensions may operate through distinct pathways and show divergent associations with vaccination outcomes; we also note cautious interpretation given multiple comparisons (Page 3-4, Lines 136-138).
- The rationale for excluding traditional media variables from regression models, despite presenting them descriptively, should be better justified analytically, because this decision may affect comparability with prior studies.
Thank you for this helpful request for clarification. The Methods (Statistical Analysis) explains that traditional media indicators were excluded from regression models because their prevalence declined substantially by 2021–22, limiting variability and interpretability in adjusted analyses; they are retained descriptively for contextual purposes (Methods 2.5, final paragraph).
- The cross-sectional nature of the regression analysis using only 2021–22 data limits causal interpretation, which should be more explicitly acknowledged in the material and methods and in the discussion.
We very appreciate for pointing this out. The limitation section acknowledges the cross-sectional design and that associations should not be interpreted causally (Limitations section, third limitation).
- The large number of stratified tables makes it difficult to identify the most policy-relevant findings. The regression results, especially the contrasting associations for measles versus OPV/DTP/PCV, are scientifically interesting but the authors should include a clearer synthesis in the text, rather than presented them primarily as statistical outputs.
Thank you for this important guidance. We revised the Results section (Section 3.4, Pages 14–15) to provide a clearer narrative synthesis of the multinomial regression findings, explicitly summarizing the direction, magnitude, and statistical significance of key associations, with particular emphasis on the contrast between measles and early multi-dose vaccines (OPV/DTP/PCV).
- Some associations appear counterintuitive (eg higher empowerment linked to lower measles completion), which calls for stronger interpretive guidance in the discussion.
Than you for noting this. We added an explicit interpretive statement at the beginning of the measles-focused discussion to frame these findings in terms of vaccine timing and the distinct structural and behavioral constraints affecting later-scheduled vaccinations (Page 18, Lines 408–411).
- The discussion is somewhat speculative regarding the underlying mechanisms. The manuscript should have a deeper engagement with health system factors such as service delivery timing, caregiver fatigue, or missed opportunities at later child ages, which could interact with empowerment in complex ways.
Thank you for this insightful suggestion. We expanded the measles-focused Discussion to incorporate health-system mechanisms from both the caregiver and supply sides, including service delivery timing constraints, cumulative burden of repeated visits, vial-opening practices for lyophilized vaccines, and missed opportunities during non-vaccination visits, supported by peer-reviewed evidence (Page 18, Lines 420–427; references [28–33]).
- As mention before .the discussion should incorporate more academic papers. The reference list relies heavily on reports and databases. The discussion would benefit from adding a small number of recent peer-reviewed studies on antigen-specific vaccination trends (Measless, DTP, ect) particularly measles, in Cambodia or the Asian region.
We very appreciate for reiterating this important point. We added six recent peer-reviewed articles to strengthen the regional and antigen-specific context, including Cambodia/Asia trend analyses and studies on missed opportunities and dropout mechanisms (new references [28–33]).
- The authors needs to better situate the findings within the broader gender and immunization literature .the discussion should incorporate more academic papers by contrasting them more explicitly with studies that report uniformly positive empowerment effects.
Thank you for this helpful recommendation. We added a concise contrast statement in the Discussion noting that much of the LMIC literature reports broadly positive associations between empowerment and immunization, whereas our antigen-specific results suggest schedule-dependent effects that may be attenuated or reversed for later-scheduled measles vaccination under time and service-delivery constraints (Page 18, Lines 411–415; citing [1,6]).
- The conclusions should be sharpened by more clearly distinguishing between empowerment as a facilitator of early-life vaccination versus a potentially insufficient factor for later-schedule vaccines like measles.
Thank you for this important point. We revised the Conclusions to explicitly distinguish the positive associations for early multi-dose vaccines (OPV/DTP/PCV) from the weaker or adverse pattern for measles at 9–12 months, and we link this to accumulated time and access constraints after mothers return to work (Page 20, Lines 514–520).
- The policy implications Should be stronger, the authors should explicitly recommended integrated strategies that combine women’s empowerment with system-level interventions targeting follow-up and retention in immunization programs.
Thank you for this constructive suggestion. We strengthened the Policy and Future Research Implications section by explicitly recommending integrated demand- and supply-side approaches, combining empowerment-oriented demand generation (health education and tailored reminders) with system-level measures to improve follow-up and retention (extended service hours, proactive defaulter tracking, and reliable outreach) to reduce missed opportunities for later-scheduled vaccines (Page 19, Lines 465–469)

Round 2
Reviewer 1 Report
Comments and Suggestions for Authors
The authors did a good job addressing the comments. I endorse it for publication.
Author Response
We sincerely thank the reviewer for the careful evaluation of our manuscript and for the positive and encouraging comments. We are pleased that the reviewer finds the manuscript suitable for publication. No further revisions were requested, and we confirm that the submitted version remains unchanged.
Thank you again for your time and support.
Reviewer 2 Report
Comments and Suggestions for Authors
The authors have addessed the concerns and I recommend the paper to be published.
Author Response

(The authors gave the same response as above.)
